Blue carbon; carbon sequestration; coastal adaptation; coastal area; coastal ecosystems

**Corresponding author:**
T. Rixen;
Email: tim.rixen@leibniz-zmt.de

# Past changes in and present status of the coastal carbon cycle

## T. Rixen 

Leibniz Centre for Tropical Marine Research (ZMT), Bremen, Germany, Institute for Geology, Universität of Hamburg, Hamburg, Germany

## Abstract

Data were obtained from the literature to identify past changes in and the present status of the coastal carbon cycle. They indicate that marine coastal ecosystems driving the coastal carbon cycle cover, on average, 5.8% of the Earth's surface and contributed 55.2% to carbon transport from the climate-active carbon cycle to the geological carbon cycle. The data suggest that humans not only increase the $CO_2$ concentration in the atmosphere but also mitigate (and before 1860 even balanced) their $CO_2$ emissions by increasing $CO_2$ storage within marine coastal ecosystems. Soil degradation in response to the expansion and intensification of agriculture is assumed to be a key process driving the enhanced $CO_2$ storage in marine coastal ecosystems because it increases the supply of lithogenic matter that is known to favour the burial of organic matter in sediments. After 1860, rising $CO_2$ concentrations in the atmosphere indicate that enhanced $CO_2$ emissions caused by land-use changes and the burning of fossil fuel disturbed what was a quasi-steady state before. Ecosystem restoration and the potential expansion of forest cover could mitigate the increase of atmospheric $CO_2$ concentrations, but this carbon sink to the atmosphere is much too weak to represent an alternative to the reduction of $CO_2$ emission in order to keep global warming below 1.5–2.°C. Although the contribution of benthic marine coastal ecosystems to the global $CO_2$ uptake potential of ecosystem restoration is only around 6%, this could be significant given national carbon budgets. However, the impact on climate is still difficult to quantify because the associated effects on $CH_4$ and $N_2O$ emissions have not been established. Addressing these uncertainties is one of the challenges faced by future research, as are related issues concerning estimates of carbon fluxes between the climate-active and the geological carbon cycle and the development of suitable methods to quantify changes in the $CO_2$ uptake of pelagic ecosystems in the ocean.

## Impact statement

Since the introduction of the term "blue carbon" more than a decade ago, coastal ecosystems have become increasingly important in the discussion on climate change mitigation and adaptation. In this work, however, I was initially interested in the role of coastal ecosystems as $CO_2$ sinks in the natural carbon cycle. It turned out that humans probably already influenced the carbon cycle in pre-industrial times by expanding agriculture. In doing so, they strengthened the function of coastal ecosystems as $CO_2$ sinks, which in turn compensated for $CO_2$ emissions caused by land-use change and brought the carbon cycle into a quasi-steady state. As a result, atmospheric $CO_2$ concentrations showed only relatively small variations. In the course of the industrial revolution, however, land-use change reached a level that almost overrode this compensation mechanism, so that it has little to counteract today's increase in atmospheric $CO_2$ concentrations due to the burning of fossil fuels. Restoration of coastal ecosystems could reinforce their $CO_2$ sink function, which could have a significant impact on reducing $CO_2$ emissions in regions where they are dominant, but globally is not an alternative to reducing $CO_2$ emissions caused by burning fossil fuel in order to keep global warming below 1.5–2°C.

## Introduction

Coasts are densely populated areas, hosting the world's largest cities, and are home to 41% of the world's population (Figure 1, Martínez et al., 2007; Small and Nicholls, 2003). In the past, a variety of diverse and closely intertwined coastal ecosystems shaped their face and acted as habitats for diverse wildlife and reaction centres that influenced the transport of carbon and other elements from land to the ocean (Cooley et al., 2022). Climate and geomorphology set the preconditions for the development of coastal ecosystems and their succession along salinity gradients. In an idealised and simplified form, the succession of coastal ecosystems on tropical deltas along estuaries starts with coastal peat swamps, which are still considered to be terrestrial ecosystems due to their low tolerance towards saltwater intrusion (Whittle and Gallego-Sala, 2016).

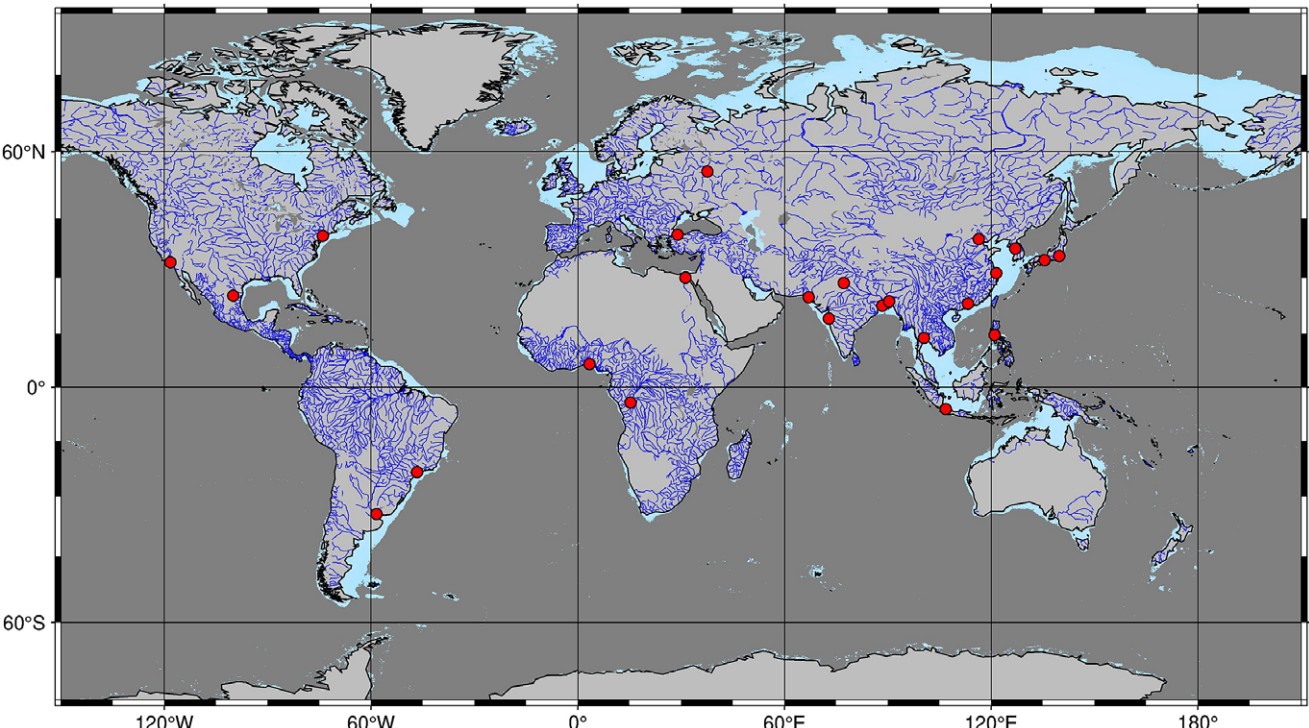

**Figure 1.** Map showing continental shelf seas (light blue, water-depth < 200 m) as well as permanent rivers (blue) and megacities (population > 15 million, red circles). The map including rivers, as well as water-depth and city population, was obtained from Uieda et al. (2022), Amante and Eakins (2009), and Demographia (2022), respectively.

In the intertidal area, brackish mangroves as well as sand- and mudflats (tidal flats) emerge, whereas permanently submerged ecosystems such as seagrass beds and coral reefs arise close to the open ocean. At higher latitudes, tidal marshes (i.e., salt marshes) displace mangroves, and cold-water corals can thrive in place of warm-water corals but in water depths of usually >40 m (Freiwald et al., 2004).

In addition to such systems, there are also coastal fog deserts and rocky shores with steeply sloping cliffs and partly adjacent beaches. In brackish fog deserts, biological soil crusts and desert plants thrive (Eckardt and Schemenauer, 1998; Martínez and Mitchell, 2017), and on rocky coasts, the steep geomorphology reduces the spatial extent of the intertidal zone and associated coastal ecosystems.

Alternatively, coastal ecosystems can be grouped into vegetated and unvegetated ecosystems (Duarte et al., 2005). Vegetated ecosystems include mangroves, salt marshes, seagrass beds and macroalgal beds such as kelp forests. Compared to angiosperm-dominated mangroves, salt marshes and seagrass beds, macroalgal beds have much greater phylogenetic diversity, and even comprise algae from different phyla (Figure 2, Krause-Jensen et al., 2018; Whittaker, 1969). In contrast to vegetated coastal ecosystems, there are also algae and bacteria that are rooted in the ground but do not form significant amounts of above-ground biomass. Ecosystems in which such organisms are the main primary producers are considered to be benthic and unvegetated coastal ecosystems. Mudflats belong to this group, and we have also included warm-water coral

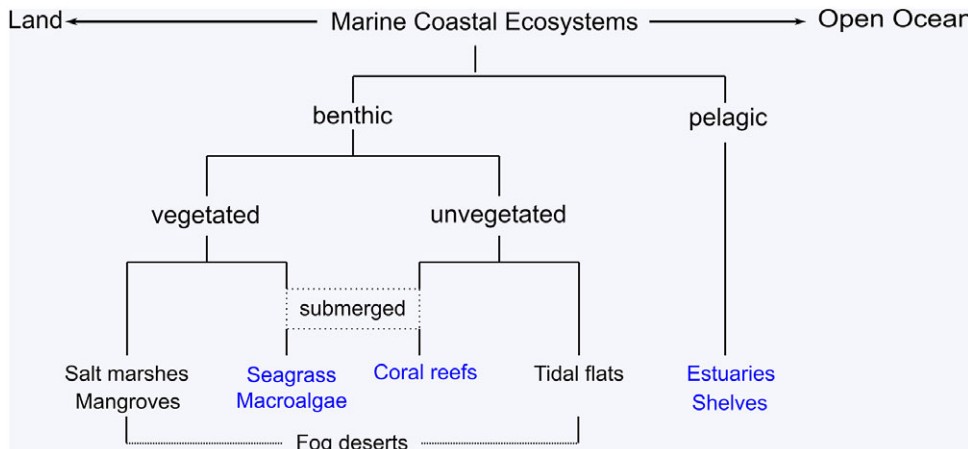

**Figure 2.** Classification of coastal ecosystems. In addition to pelagic ecosystems, also submerged ecosystems which thrive below the sea surface are considered to be part of the ocean (blue). Non-submerged benthic ecosystems are considered to be part of the land surface (black).

reefs, although the algae in these reefs do not thrive freely but predominantly as coral symbionts. Since cold-water corals grow mainly in greater water depths and thus in the dark, they do not harbour photoautrophic but chemoautrophic algal symbionts (Middelburg et al., 2015). Therefore, we do not regard it as a typical coastal ecosystem. Nevertheless, half of the cold-water reef area was considered a submerged coastal ecosystem thriving on shelves, while the other half was considered part of the open ocean.

In fog deserts, the availability of water creates a gradient from bare sand deposits to ecosystems dominated by biological soil crusts and shrubs. Because (in contrast with bare sand deposits) above-ground biomass shapes the appearance of shrublands, fog deserts can change from unvegetated to vegetated coastal ecosystems. A lack of information on the extent of fog deserts means that they will not be included in the following discussion. However, phytoplankton, including eukaryotic algae and bacteria, is the main primary producer in pelagic coastal ecosystems thriving in estuaries and on shelves (Falkowski et al., 2004).

### The global coastline and spatial extent

Coastal cliffs and sandy beaches occur along approximately 52% (Young and Carilli, 2019) and 31% of the world's shorelines (Luijendijk et al., 2018). The term rocky shore can, in turn, refer to coastal cliff sections that are also foregrounded by sandy beaches. Hence, rocky shores (with and without sandy beaches) and land–sea interfaces covered only by flat-lying soft sediment (i.e., mud coasts) account for 52% and 48% of the world's coastline, respectively. Macroalgae and seagrass beds are assumed to dominate along rocky shores, whereas peat swamps, tidal flats, mangroves and salt marshes often characterise muddy coasts (Duarte, 2017; Young and Carilli, 2019).

The spatial extent of ecosystems and specific land covers has been obtained from the literature and is presented in Table S1 in the Supplementary Material including references. In order to avoid double counting and to distinguish between the terrestrial biosphere (land), the open ocean and marine coastal ecosystems, the spatial extents have partly been recalculated (see Table S2 in the Supplementary Material). For example, since submerged coastal ecosystems and estuaries are part of the continental shelf, their areas were subtracted from the global shelf extent (26.39–32.24 $10^6$ km$^2$) to obtain the area of the open shelf. The global shelf area was subtracted from the area of the global ocean (361.88 $10^6$ km$^2$) to estimate the area of the open ocean (329.5–335.36 $10^6$ km$^2$) including half of the cold-water coral reef area. Land, in turn, comprises urban areas and regions covered by mineral soils, ice and wetlands, while wetlands include non-submerged coastal ecosystems (mangroves, salt marshes and mudflats), peat swamps and inland waters. To distinguish between land, open ocean and marine coastal ecosystems, non-submerged coastal ecosystems have been subtracted from the land area and added to the area of submerged and pelagic coastal ecosystems. On average, this means that marine coastal marine ecosystems cover about 5.8% of the Earth's surface, while the remaining part is occupied by the open ocean (65.2%) and land (29.0%).

### Carbon cycles

Marine coastal ecosystems cover only a relatively small fraction of the Earth's surface (5.8%), but they are known to be of great relevance for the global carbon cycle and atmospheric $CO_2$ concentrations due to their high carbon turnover rates (e.g., Ciais et al., 2013; Regnier et al., 2013; Najjar et al., 2018; Wit et al., 2018). Hence, marine coastal ecosystems gain a socioeconomic and political dimension, which in the current climate discussion is strongly associated with the term "blue carbon" (Nellemann et al., 2009; Lovelock and Duarte, 2019). Before discussing this concept, we will introduce the global carbon cycle because it is the conceptual framework on which our understanding of the role of coastal ecosystems in the climate system is based. The global carbon cycle began to be mentioned in the years between 1840 and 1845 (Galvez and Gaillardet, 2012), and gained momentum as the greater availability of computers favoured the development of numerical models in the 1980s and 1990s (e.g., Walker et al., 1981; Broecker and Peng, 1982; Volk and Hoffert, 1985; Berner, 1991). It still poses a number of open questions (e.g., Lee et al., 2020; Müller et al., 2022), but the discussion of which goes beyond the scope of this article.

In principle, the global carbon cycle can be divided into a geological and a climate-active carbon cycle (Figure 4, Berner and Caldeira, 1997; Ridgwell and Zeebe, 2005). The geological carbon cycle describes the cycle of carbon within the interior of the Earth that influences the Earth's climate on geological time scales. The climate-active carbon cycle operates at the Earth's surface and largely controls the contemporary climate. It comprises three carbon reservoirs: the ocean, the atmosphere and the terrestrial biosphere. Lateral carbon fluxes between the ocean and the terrestrial biosphere as well as vertical carbon fluxes between these two reservoirs and the atmosphere control the cycling of carbon within the climate-active carbon. Since this work focuses mainly on processes that control the total amount of carbon within the climate-active carbon cycle and the resulting effects on atmospheric $CO_2$ concentration, it neglects lateral fluxes and concentrates on processes that link the geological and climate-active carbon cycles. They include carbon sources and carbon sinks (Table 1). Carbon sources deliver gaseous $CO_2$ and dissolved inorganic carbon (DIC) from the geological to the climate-active carbon cycle, while carbon

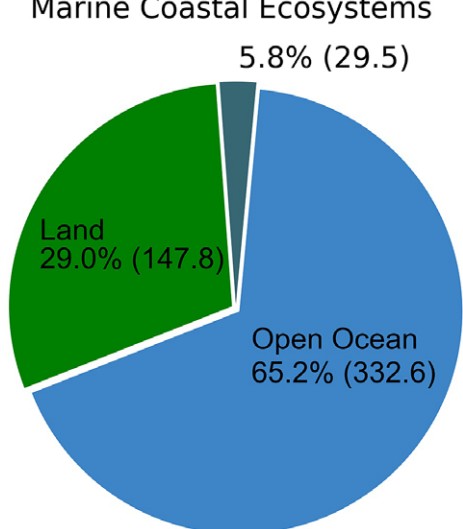

**Marine Coastal Ecosystems**
5.8% (29.5)

Land 29.0% (147.8)

Open Ocean 65.2% (332.6)

numbers in brackets are areas in $10^6$ km$^2$

**Figure 3.** Distribution of marine coastal ecosystems, land (excluding the spatial extent of non-submerged ecosystems) and the open ocean on the Earth's surface (references are given in Table S1 and S2 in the Supplementary Material).

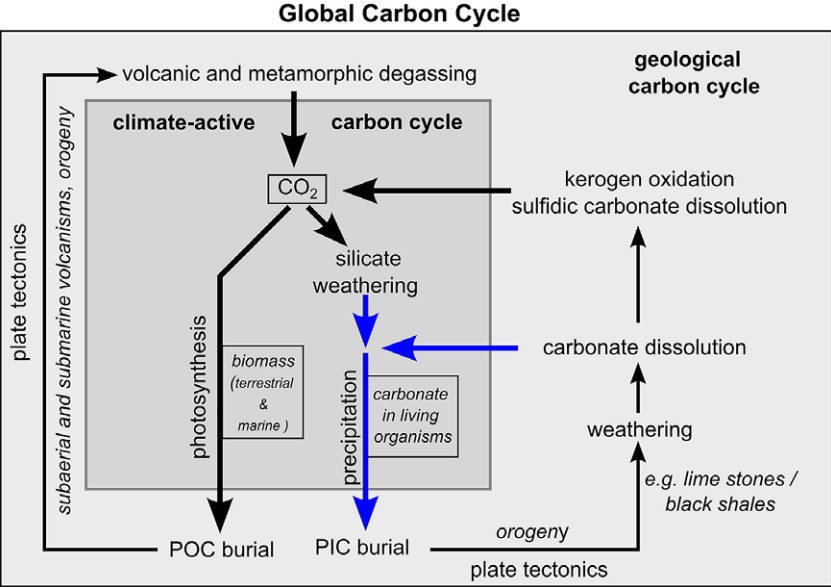

**Figure 4.** Schematic illustration of the global carbon cycle and processes linking the geological and the climate-active carbon cycles. The blue arrows represent the supply and removal of dissolved carbonate through weathering the burial of PIC, respectively. The boxes ($CO_2$, biomass and carbonate) represent carbon reservoirs. "$CO_2$" represents atmospheric $CO_2$ and $CO_2$ dissolved in the ocean and freshwater on land. "biomass" and "carbonate" represent the storage of POC and PIC in living organisms.

sinks remove carbon in the form of particulate organic carbon (POC) and inorganic carbon (PIC) from the climate-active carbon cycle.

The burial of PIC and POC in sediments and soils is a carbon sink for the climate-active carbon cycle, and the contribution of marine coastal ecosystems to the total burial of PIC and POC is a measure to determine their importance for the climate-active carbon cycle. The fixation of $CO_2$ in biomass through photosynthesis and the carbonate precipitation are prerequisites for the burial of POC and PIC. However, only a minor fraction of the PIC and POC produced reaches the sediments and soils. In the ocean and underlying sediment, most of the POC and PIC produced are remineralised and dissolved, respectively, in seawater and pore water (e.g., LaRowe et al., 2020; Sulpis et al., 2021). In anoxic sediments, methane produced by anaerobic degradation of POC can be trapped in gas hydrates and authigenic carbonates (Wallmann and Aloisi, 2012). Accordingly, the storage of carbon in gas hydrates and authigenic carbonates is considered as much a carbon sink as the burial of POC. Nevertheless, not all the POC stored in sediment and soils is produced in today's climate-active carbon cycle. Rock organic matter such as kerogen, for instance, is POC that was produced in the geological past. Black shales, for example, owe their colour to it. It can be oxidised, but also physically eroded, transported and buried in sediments without significant chemical alteration (Blattmann, 2022). Therefore, the burial rates of POC must also be corrected for the burial of kerogen before they represent a $CO_2$ sink for the climate-active carbon cycle. In addition, the accumulation of POC in soils on land and also the storage of respired POC in deep groundwater aquifers have been suggested as $CO_2$ sinks. Since this process is still poorly quantified (Naorem et al., 2022 and references therein), it was neglected. Furthermore, PIC burial rates in soils can be extremely high with values of >4,000 g C m$^{-2}$ yr$^{-1}$ (e.g., Zamanian et al., 2016), but it was set to zero (Table 1) because it is mostly driven by the precipitation of dissolved carbonates provided by the dissolution of carbonate rocks, which cover ~20% of the Earth's land surface (Hartmann and Moosdorf, 2012).

Volcanic and metamorphic degassing of $CO_2$ as well as weathering act as carbon sources for the climate-active carbon cycle. The oxidation of rock organic matter and the dissolution of carbonate minerals are weathering immanent carbon sources. Similar to the volcanic and metamorphic degassing, they introduce carbon as gaseous $CO_2$ (kerogen oxidation) but also as DIC (dissolution of carbonate minerals) into the climate-active carbon cycle. However, carbonate dissolution can also serve as a direct source of $CO_2$ if carbonate rocks (e.g., lime stones and marble) contain significant amounts of sulphides such as pyrite. Under such conditions, pyrite oxidation lowers the pH, which favours the formation and outgassing of $CO_2$ from the DIC pool (Lerman et al., 2007; Bufe et al., 2021). This process is related to the carbonate system described in the next section: "Partitioning of carbon between the ocean and the atmosphere".

Silicate weathering on land and in the ocean, where, for example, oceanic crust is exposed to seawater, differs from the weathering processes discussed so far. Instead of dissolving carbonate minerals that are part of the geological carbon cycle, $CO_2$ that is already part of the climatic carbon cycle (i.e., atmospheric $CO_2$ and marine DIC) is converted into dissolved carbonate, as described in general terms in the Urey equation (Urey, 1952) and more specifically by, for example, Hartmann et al. (2013). The release of dissolved carbonate by the dissolution of carbonate minerals and silicate weathering as well as the precipitation of dissolved carbonate and its burial as PIC can be considered as the carbonate cycle, which strongly influences carbon partitioning between the ocean and the atmosphere (Figure 4).

## Partitioning of carbon between the ocean and the atmosphere

The balance between the dissolved carbonate supply to the ocean and the marine PIC burial affects the distribution of carbon between the ocean and the atmosphere through its influence on the ocean's pH and the marine carbonate system (Broecker, 1983; Zeebe and Wolf-Gladrow, 2001; Sarmiento and Gruber, 2006). The

**Table 1.** Carbon fluxes into and out of the climate-active carbon cycle and transformation processes acting within the climate-active carbon cycle

| | Tg C yr.$^{-1}$ | | |
| | Mean | ± | References |
|---|---|---|---|
| **Carbon sources** | | | |
| Degassing from the solid Earth | | | |
| Subaerial volcanism | 85.3 | 17.1 | (Werner et al., 2020) |
| Submarine volcanism | 37.2 | 22.8 | (Isson et al., 2020) |
| Metamorphic degassing | 39.9 | 8.1 | (Isson et al., 2020) |
| Weathering | | | |
| Carbonate dissolution | 154.9 | 13.2 | (Gaillardet et al., 1999)[a] |
| Sulfidic carbonate diss. | 255.0 | 69.1 | (= terr. Silicate weathering)[b] |
| Organic carbon oxidation | 117.6 | 74.6 | (Kerrick and Caldeira, 1998; Wallmann and Aloisi, 2012) |
| **Total carbon source** | **690.0** | **204.8** | |
| **Carbon sinks** | | | |
| Carbonate carbon (PIC) burial | | | |
| Soil | 0 | 0 | |
| Sediment | **563**.0 | 24**3.6** | |
| Alteration of oceanic crust | 23.4 | 21.9 | (Wallmann and Aloisi, 2012; Coogan and Gillis, 2013) |
| PIC total | 58**6**.6 | 26**5**.5 | |
| Organic carbon (POC) burial | | | |
| Soil | 21**4.7** | 59.**3** | |
| Sediment | 34**3**.5 | 19**5**.6 | |
| Subtotal | 508.2 | 214.7 | |
| Authigenic carbonate | 12.0 | 6.0 | (Sun and Turchyn, 2014) |
| Gas hydrates | 4.3 | 3.0 | (Wallmann and Aloisi, 2012) |
| Kerogen burial | −60.0 | 40.0 | (Blattmann, 2022) |
| POC total | 514.4 | 303.9 | |
| **Total carbon sink** | **1,100.1** | **569.4** | |
| **Transformation processes: $CO_2 < - > HCO_3^-$** | | | |
| Terrestrial silicate weathering | 255.0 | 69.1 | (Gaillardet et al., 1999; Hartmann et al., 2009)[a] |
| Marine silicate weathering | 197.3 | 133.3 | (Isson et al., 2020)[c] |
| Reversed weathering | −63.1 | 57.1 | (Isson et al., 2020)[d] |
| **Net silicate weathering** | **389.2** | **259.4** | **(terr. + mar.) – (reversed weathering)** |

[a]Including the data listed by Gaillardet et al. (1999) in their Table 4.
[b]This estimate is based on results obtained by Bufe et al. (2021) showing that $CO_2$ emission caused by sulfidic carbonate dissolution could be in the same range or even exceed $CO_2$ uptake by terrestrial silicate weathering.
[c]This estimate includes the marine silicate weathering of $173.7 \pm 114.4$ TgCyr$^{-1}$ and the alteration of the oceanic crust of $23.4 \pm 21.9$ Tg C yr.$^{-1}$, as derived from Wallmann and Aloisi (2012).
[d]This refers to authigenic clay production, which reconverts dissolved carbonates into $CO_2$ (Isson et al., 2020).

marine carbonate system is a series of equilibrium reactions controlling the distribution of carbon species ($CO_2$, $HCO_3^-$ and $CO_3^{2-}$) within the DIC and therewith fluxes of $CO_2$ across the air–water interface (Figure 5). The carbonate system shifts towards $CO_2$ and $CO_3^{2-}$ if the pH decreases and increases, respectively. The first favours $CO_2$ emission into the atmosphere because it enhances the $CO_2$ concentration in ocean waters. The latter increases the $CO_2$ uptake of the ocean as it lowers the $CO_2$ concentrations in ocean waters. The dissolution and supply of dissolved carbonate ($CaCO_3$ - > $Ca^{2+}$ + $CO_3^{2-}$) increases the pH value because it provides $CO_3^{2-}$. This molecule is twofold negatively charged but carries only one mole of carbon. Therefore, it increases the total alkalinity (TA) by two units and the DIC by only one unit, with the result that the positive influence of the TA overcompensates the negative effects of the DIC-increase on the pH value (Zeebe and Wolf-Gladrow, 2001). The precipitation of carbonate has an opposing effect on the pH. Accordingly, an excess supply of dissolved carbonate from weathering favours the $CO_2$ uptake of the ocean by increasing the pH, whereas an excess removal of DIC and TA through the precipitation of carbonate and the subsequent PIC burial decreases the carbon storage in the ocean by lowering the pH (Broecker, 1983).

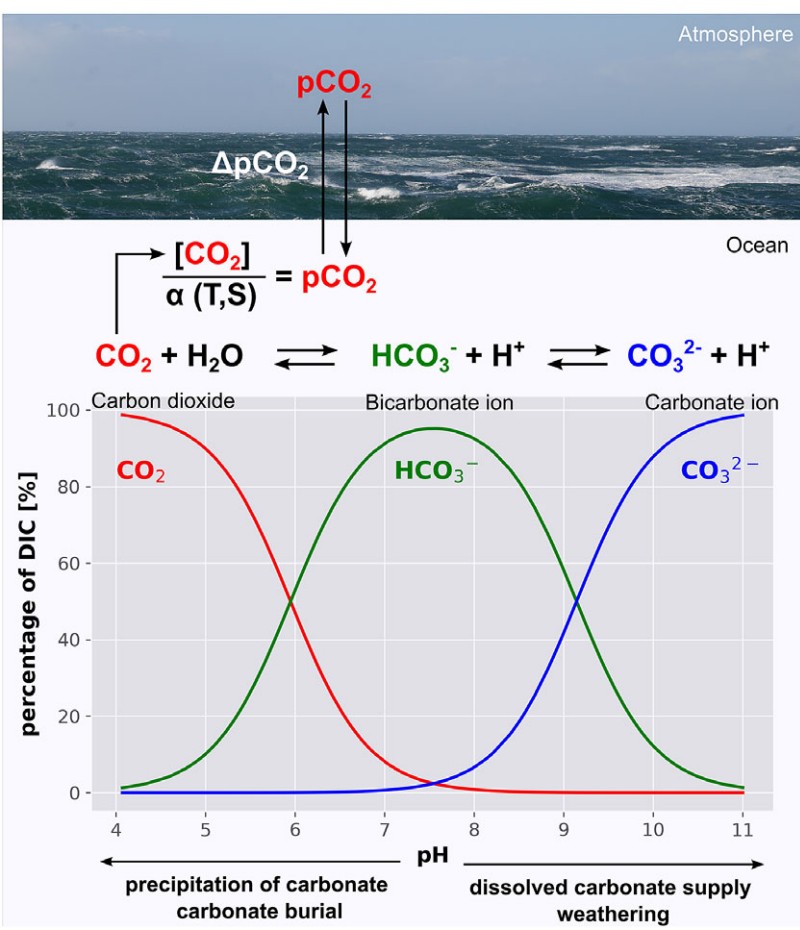

**Figure 5.** Schematic illustration of the marine carbonate system and its influence on the $pCO_2$, and a Bjerrum plot showing pH effects on the contribution of the individual carbonate species ($CO_2$, $HCO_3^-$ and $CO_3^{2-}$) to the total dissolved inorganic carbon (DIC). Arrows at the bottom indicate the effects of calcification of the subsequent burial of carbonate minerals and the weathering supply of dissolved carbonate on the carbonate system. The $pCO_2$ is the quotient of the $CO_2$ concentration and its solubility ($\alpha$), which in turn depends on the seawater temperature (T) and salinity (S). The differences between the $pCO_2$ in the atmosphere and ocean ($\Delta pCO_2$) determine the direction of $CO_2$ fluxes. A $pCO_2$ in the ocean exceeding those in the atmosphere causes $CO_2$ emissions and vice versa; a lower $pCO_2$ leads to the transfer of $CO_2$ from the atmosphere into the ocean.

## Steady-state and human perturbations

Constant $CO_2$ concentration in the atmosphere represents, in general, a climate-active carbon cycle at a steady state which means that carbon inputs equal carbon outputs. After the last glacial maximum at ~18,000 years before present (BP, present = 1950) the $CO_2$ concentration in the atmosphere increased and reached a plateau with a relatively low variability only ca. 2,500 to 2,000 years before present (e.g., Indermuhle et al., 1999; Schmitt et al., 2012, Figure 6). Before that time and in the aftermath of the deglaciation, the adaptation of ecosystems to global warming strongly influenced the atmospheric $CO_2$ concentrations. The re-growing terrestrial biosphere acted as a $CO_2$ sink for the atmosphere and caused negative feedback on the marine carbon cycle which turned into a $CO_2$ source for the atmosphere (e.g., Broecker et al., 2001; Brovkin et al., 2016). With the stabilisation of the post-glacial sea level rise at around 8,200 BP (Fairbanks, 1989; Gischler et al., 2008; Lambeck et al., 2014), today's coastal ecosystems evolved and the expansion of coral reefs is assumed to have contributed to the ~20 ppm increase of the atmospheric $CO_2$ concentrations between approximately 7,500 and 2,500 BP (e.g., Ridgwell et al., 2003).

However, despite opposing opinions, there seems to be a consensus that $CO_2$ emissions caused by human-induced land-use changes were, with 30–100 TgCyr$^{-1}$ (Pongratz et al., 2009; Bauska

et al., 2015), too low to have influenced $CO_2$ concentrations in the atmosphere significantly during pre-industrial times (Brovkin et al., 2016; Ruddiman et al., 2016). Furthermore, during the last 1,000 years, the carbon isotopic composition of $CO_2$ in the atmosphere, along with varying $CO_2$ concentrations, indicates that $CO_2$ emissions caused by land-use changes could have also been balanced by a [13]C-depleted carbon sink (Bauska et al., 2015). Although this sink is not specified, such a balance, along with a comparably stable concentration of atmospheric $CO_2$, supports the view that the climate-active carbon cycle was in quasi-steady state during the last 1,000 years until 1860 (i.e., before the industrial age). Nevertheless, the climate was not constant. It revealed, for example, a long-term cooling trend between ~1580 and 1880 (Wang et al., 2009; Büntgen et al., 2011; Ahn et al., 2012; PAGES 2k Consortium, 2013; Bauska et al., 2015; Goosse et al., 2022).

After 1860, the burning of fossil fuels to power the industrial revolution brought the quasi-steady state to an end, as was indicated by increasing concentrations of atmospheric $CO_2$ (Figure 6). The intensification of agriculture in response to population growth and the discovery of fertiliser between ~1840 and 1910 accompanied this transition and led to $CO_2$ emissions caused by land-use changes of 670 Tg Cyr$^{-1}$ in 1860 (Friedlingstein et al., 2020). These perturbations accelerated after 1950 (Steffen et al., 2015), and today, the fight against raising $CO_2$ levels in the atmosphere and the

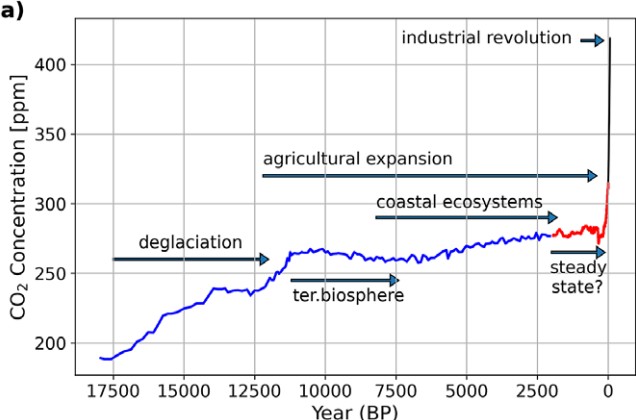

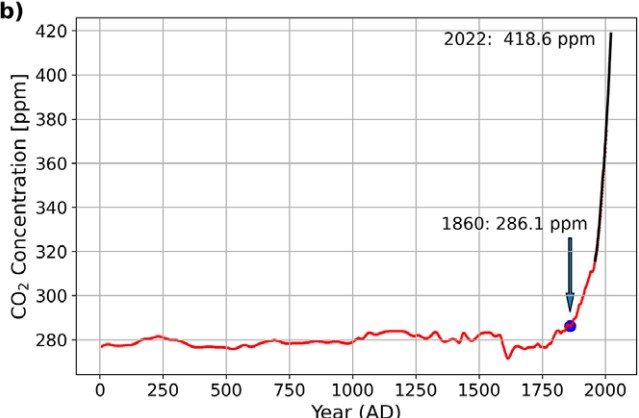

**Figure 6.** Atmospheric CO$_2$ concentrations during the last ~18,000 years (a) and 2,000 years (b). Data marked in blue (Monnin et al., 2001) and red (MacFarling Meure et al., 2006) are derived from ice cores, while the colour black indicates annual mean Mauna Loa data (Keeling, 1960). The Mauna Loa data were downloaded in August 2022 from https://gml.noaa.gov/webdata/ccgg/trends/co2/co2_mm_mlo.txt. The text and arrows indicate major natural and human effects on the climate-active carbon cycle. In (a), "terrestrial biosphere" marks the period as the regrowth of the terrestrial biosphere decreased the atmospheric CO$_2$ concentrations. "Coastal ecosystem" indicates the time as the stabilising post-glacial sea level rise allowed the development of the contemporary coastal ecosystems and "agricultural expansion" marks the time since the origin of agriculture in the fertile crescent. In (c) the numbers indicate atmospheric CO$_2$ concentration in years 1860 and 2021.

resultant global warming is considered one of the greatest challenges facing humanity (Editorial, 2019).

## Blue carbon

The 2015 United Nations Climate Change Conference (COP 21) in Paris agreed to reduce CO$_2$ emissions and keep global warming below 1.5–2 °C (Figueres et al., 2017). Inspired by the colour of the ocean and the intention to include marine and coastal ecosystems in climate change mitigation and adaptation strategies, Nellemann et al. (2009) introduced the blue carbon concept. In addition to reliable accounting of POC storage within blue carbon ecosystems, the blue carbon concept encompasses socially acceptable management strategies that are in alignment with other mitigation and adaptation approaches (Lovelock and Duarte, 2019).

Lovelock and Duarte (2019) distinguished between well-established blue carbon ecosystems (mangroves, salt marshes and seagrass), emerging blue carbon ecosystems such as tidal flats and macroalgae, and coastal ecosystems that did not meet the criteria of a blue carbon ecosystem. Coral reefs were grouped into the latter

category because instead of high POC burial rates, coral reefs are characterised by high rates of carbonate precipitation and PIC burial which can increase CO$_2$ concentrations in the atmosphere by lowering the pH in seawater, as was discussed above (Figure 5). Recent studies have also raised the question as to whether PIC burial might counteract CO$_2$ sequestration in seagrass beds, which is a well-established blue carbon ecosystem (Mazarrasa et al., 2015; Macreadie et al., 2017). However, whether PIC burial acts as a CO$_2$ sink or source for the atmosphere depends on the state of the climate-active cycle and the integrated carbonate cycle (Figure 4.). To obtain an assessment of the state of the climate-active carbon cycle and the importance of marine coastal ecosystem for the climate-active carbon cycle, we compiled burial rates of POC and PIC from the literature (Figure 7) and compared them with carbon inputs to the climate-active carbon cycle, as described in the section "Blue Carbon". However, the uncertainties associated with these estimates are substantial (Williamson and Gattuso, 2022 and references therein). Notwithstanding the resulting issues regarding statistical significance, such estimates were the only way we could establish the possible state of the climate-active carbon cycle and the potential role coastal ecosystems play within it.

## Particulate organic carbon burial rates

Burial rates compiled from the literature are assumed to characterise those of largely undisturbed ecosystems. The burial rates are given as area-normalised flux densities (Figure 7a,c), expressing burial efficiency, and POC and PIC burial rates (Figure 7b,d). The latter are the product of the flux density and the spatial extent (see Table S2 in the Supplementary Material) of the respective ecosystem.

High POC burial efficiency characterises conventional blue carbon ecosystems (i.e., mangroves, salt marshes and seagrass) and tidal flats; shelves have the highest POC burial rates because of their comparably large spatial extent (Figure 7). Macroalgae beds have a low POC burial efficiency and consequently a low POC burial rate because they are predominant on rocky shores. In comparison with muddy coasts, the accumulation of sediment is generally low at such sites; it is difficult to store organic matter in the absence of sediment.

Rock weathering and (in marine systems) bio-mineralisation, such as calcification, supply the minerals in soils and sediment, while small rock fragments are often referred to as lithogenic matter. The accumulation of bio-minerals and lithogenic matter favours the POC burial by building a habitat and increasing the preservation of organic matter. Large amounts of lithogenic matter can increase POC burial rates by reducing its residence time in the water column and surface sediment, where biological activity and thus remineralisation rates are high. Furthermore, the adsorption and/or integration of organic molecules onto and into mineral structures protects organic matter against bacterial attack (Keil and Hedges, 1993; Hedges and Keil, 1995; Armstrong et al., 2002; Rixen et al., 2019; Georgiou et al., 2022). Since in marine coastal ecosystems only <16% of the primary produced organic matter is buried in sediments (Emerson and Hedges, 1988; Duarte, 2017), the supply of minerals and their effects on remineralisation and burial of POC – rather than primary production – can be considered as the prime factor influencing POC burial.

However, POC buried in sediment is not only produced within the respective ecosystem (= autochthonous POC) but also imported from other ecosystems (= allochthonous POC). In mangroves (24–80%), seagrass beds (70–90%) and salt marshes (11–78%), the contribution of allochthonous POC is extremely variable

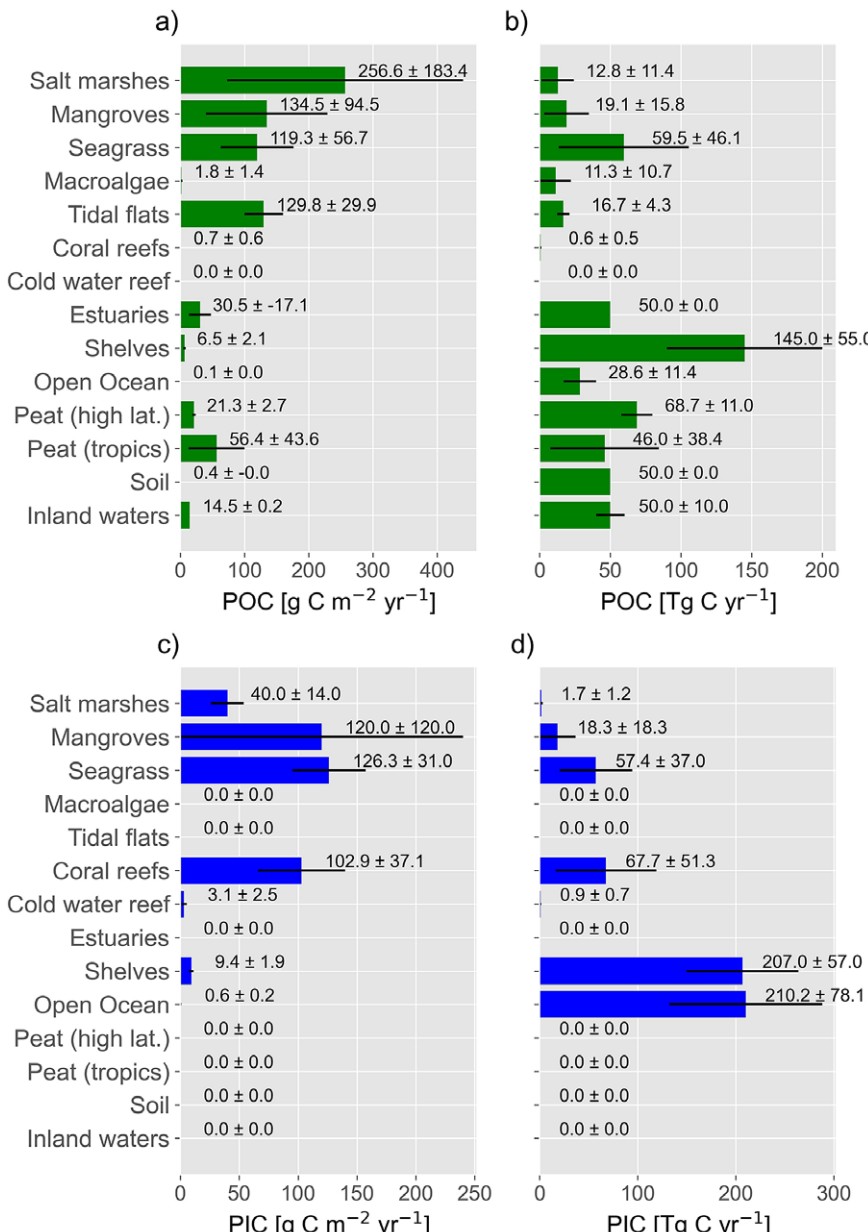

**Figure 7.** POC (a) and PIC (c) carbon burial densities as well as POC (b) and PIC (d) burial rates (d), as derived from data shown in Table S3. Table S3 in the Supplementary Material also shows the respective references.

(Williamson and Gattuso, 2022). In contrast, macroalgae growing on rocky shores often act as a source of allochthonous POC because POC produced by macroalgae is carried offshore and buried in marine sediment (Duarte et al., 2013). Overall, biogeochemical tracers indicate that approximately one-third of the POC buried in marine sediment originates from terrestrial plants (Burdige, 2005). Hence, marine ecosystems act as depocenters for autochthonous and allochthonous POC. According to the compiled data, POC burial in marine coastal ecosystems (autochthonous and allochthonous POC) amounts to $314.9 \pm 143.9$ Tg yr.$^{-1}$ contributing, on average, 56.4% to the global mean POC burial rate of $558.2 \pm 214.7$ Tg yr.$^{-1}$ (Figure 8a). POC burial in marine coastal ecosystems and the open ocean amounts to 343.5 Tg Cyr-1 (see Table 1).

### Particulate inorganic carbon burial rates

Pelagic ecosystems in the open ocean, continental shelf seas, as well as coral reefs and seagrass beds show the highest PIC burial rates (Figure 7d). Since calcifying organisms thrive mainly in pelagic ecosystems and coral reefs (Wilson et al., 2009; Lebrato et al., 2010; Lebrato et al., 2016), this result was expected, but the high PIC burial rate in seagrass was not. Although the seagrass *Thalassia testudinum* can accumulate carbonate crystals within its cell walls and externally on its surface (Enríquez and Schubert, 2014), epiphytes (along with pelagic and benthic calcifiers) are assumed to be the main carbonate producers in seagrass beds (Mazarrasa et al., 2017). Lateral transportation of carbonates from nearby reefs may contribute to high PIC burial rates in tropical seagrass beds, while

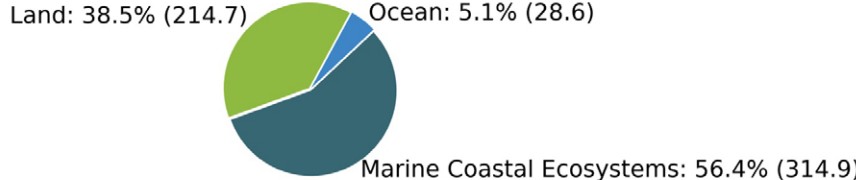

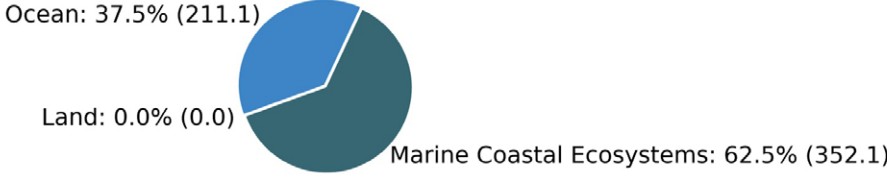

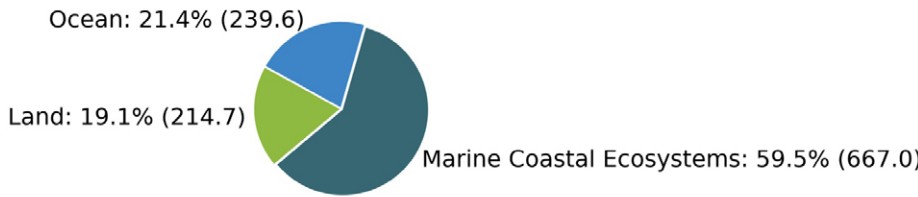

numbers in brackets are fluxes in TgCyr$^{-1}$

**Figure 8.** POC (a), PIC (b) and total carbon burial rates (c).

acidic conditions prevent PIC burial in peat swamps. Since estimates on global PIC burial rates in tidal flats, macroalgae beds, inland waters and soils are not available, they were set to zero (Figure 7b). Hence, PIC burial rates were underestimated, but based on the available data it amounts to 352.1 ± 164.8 TgCyr$^{-1}$. Therewith, marine coastal ecosystems contribute 62.5% to global PIC burial rates (Figure 8b) and 59.5% to the total carbon (PIC and POC) burial (Figure 8c).

Considering a mean kerogen burial of 60 Tg kerogen C yr-1 (see Table 1) and assuming that this burial occurs in marine coastal ecosystems reduces the respective total PIC and POC burial from 667.0 TgCyr-1 (Figure 8c) to 607 TgCyr-1. In comparison to the total carbon transfer from the climate active to geological carbon cycle of 1100.1 TgCyr-1 (Table 1), this indicates that marine coastal ecosystems are responsible for 55.2% of the total carbon transport from the climate-active carbon cycle to the geological carbon cycle. Hence, they form a crucial part of the climate-active and the geological carbon cycles even though they cover only 5.8% of the Earth's surface.

## State of the climate-active carbon cycle

Since climate-active carbon cycle was in quasi-steady state until 1860 and the compiled POC and PIC burial rates are assumed to characterise those of largely undisturbed ecosystems, it is expected that (1) PIC and POC burial equals carbon inputs into the climate activate carbon cycle and (2) PIC burial balances inputs of dissolved carbonate released during the dissolution of carbonate rocks and the silicate weathering.

## Carbonate cycle

According to the compiled data (Table 1), PIC burial rates including the alteration of the ocean crust are within the same range, as estimates for the release of DIC by the dissolution of carbonate minerals and silicate weathering, as expected for a steady state (Figure 9). A PIC burial that equals the supply of dissolved carbonate implies, in turn, that the PIC burial is a $CO_2$ sink for the

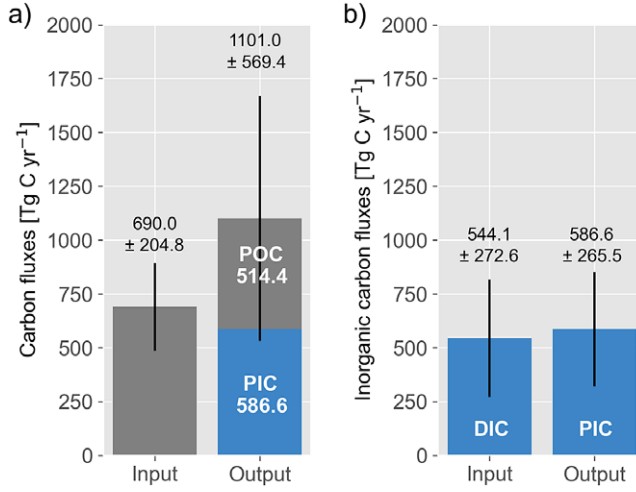

**Figure 9.** Carbon inputs via weathering and degassing and outputs via PIC (blue) and POC (grey) burial into and from the climate-active carbon cycle (a). Bicarbonate input via weathering and output via PIC burial (b).

atmosphere because it removes carbon from the climate-active carbon cycle, which silicate weathering and the dissolution of carbonate had formerly converted into dissolved carbonate without changing the ocean's pH. Accordingly, coral reefs might also meet the criteria of a blue carbon ecosystem albeit with a significant difference regarding ecosystem degradation. Ecosystem degradation generally lowers the $CO_2$ storage in well-developed and emerging blue carbon ecosystems because it reduces biomass and POC burial. In coral reefs, POC storage and burial are of minor importance, while a reduced coral growth and dissolution of reef carbonates favour the $CO_2$ uptake of the ocean by enhancing the supply of dissolved carbonate and reducing the PIC burial.

Since 1860, rising atmospheric $CO_2$ concentrations and the release of NOx, due, for example, to the use of fertilisers, have acidified the ocean and soils (Kleypas et al., 2006; Shi et al., 2012; Zamanian et al., 2016) and potentially disturbed this steady state. Ocean acidification hinders the growth of calcifying organisms (Pörtner et al., 2014; Davis et al., 2021) and leads to the dissolution of reef carbonates (Eyre et al., 2018), while soil acidification is assumed to enhance the dissolution of soil carbonates and concrete in urban areas (Washbourne et al., 2015; Zamanian et al., 2016; Naorem et al., 2022). Hence, an increased dissolved carbonate supply due to soil acidification and a reduced PIC burial as a result of ocean acidification could therefore strengthen the ocean's role as an sink of atmospheric. However, the near-balanced input and removal of inorganic carbon from climate-active carbon (Figure 9b) does not yet support such a trend.

### Climate-active carbon cycle

In contrast with the carbonate cycle, the climate-active carbon cycle reveals a mean carbon input that falls below the output (Figure 9a). In this case, the difference between the mean input and output is so large that it merits further discussion despite the large error ranges because it sheds an interesting light on the human impact on the climate-active carbon cycle.

Before looking at human influence, it is important to state that it is negligible on decadal-to-centennial timescales in the case of volcanic and metamorphic degassing and the dynamic of $CH_4$ in deep-sea sediment. Furthermore, the nearly balanced carbonate cycle suggests that human influence on weathering and PIC burial was neglectable at least before ~1860. Therefore, increased POC burial rates must have caused the disequilibrium between carbon inputs and outputs. This runs counter to the quasi-steady state that prevailed during the 1,000 years before 1860 (Figure 6b). To sustain this state, a POC burial rate of 103.4 TgC yr$^{-1}$ would have been sufficient to balance carbon inputs (Figure 10a). Accordingly, the difference between a steady-state carbon input of 690.0 TgC yr$^{-1}$ and an output of 1,101.0 TgC yr$^1$ indicates an excess POC burial rate of 411.0 TgC yr$^{-1}$.

The excess POC burial rate represents a $CO_2$ sink for the atmosphere, and the stored carbon is $^{13}$C-depleted due to the use of the enzyme RuBisCO (Ribulose-1,5-bisphosphate carboxylase) to fix $CO_2$ during the photosynthesis and the associated isotopic fractionation (Schidlowski, 1988; Hayes, 1993). Such a $^{13}$C-depleted $CO_2$ sink would have been sufficient to balance $CO_2$ emission caused by land-use changes of 30–100 TgCyr$^{-1}$ and to establish the quasi-stead state before 1860. Agricultural practices are assumed to have been the driving forces because of two main reasons: First, they lead to soil erosion and enhanced lithogenic matter supply (van Andel et al., 1990; Syvitski and Vörösmarty, 2005; Mulitza et al., 2010; Burdanowitz et al., 2021), and second,

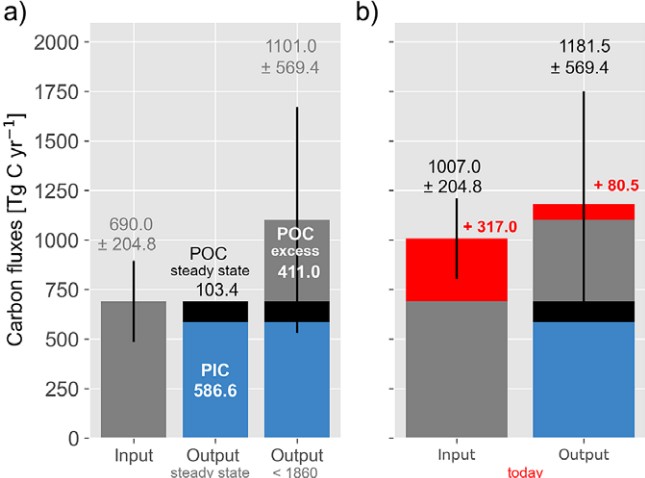

**Figure 10.** (a) Carbon input and output as shown in Figure 9 but including a steady state (s.s.) output that may have prevailed during pre-industrial background conditions. (b) Carbon input and output today, where soil erosion enhances the carbon input (red) and output (red). Carbon inputs through $CO_2$ emissions from fossil fuel combustion and net cement production as well as land-use changes have not been included.

marine coastal ecosystems trap the vast majority of the eroded lithogenic matter (> 80%, Hedges and Keil, 1995; McKee et al., 2004). Accordingly, agriculture appears to not only be the driving force behind $CO_2$ emissions caused by land-use change but also act as a $CO_2$ sink by providing lithogenic material and promoting the burial of POC in marine coastal ecosystems. The Neolithic agricultural revolution (~12,200–11,700 BP) might have initiated this compensatory mechanism, which gained momentum with the subsequent spread of agriculture.

Today, $CO_2$ emissions from land-use change, fossil fuel combustion and net cement production amount to about 10,000 TgCyr$^{-1}$ and exceed previously considered carbon fluxes by an order of magnitude (Friedlingstein et al., 2020). In addition to these overwhelming perturbations of the climate-active carbon cycle, the PIC and POC burial rates have also been changed. According to recent estimates, POC burial rates in inland waters and in marine coastal ecosystems have increased by 100 TgC yr$^{-1}$ and 50 TgC yr$^{-1}$ respectively, (Regnier et al., 2022). At the same time, the accumulation of eroded soils in reservoirs has reduced the global input of lithogenic matter to the ocean by about 10% (Syvitski and Vörösmarty, 2005). However, in addition, soil erosion and degradation appear to have transformed soils from carbon sinks to carbon sources. The net loss of POC from mineral soils (globally) and peat soil in Indonesia, where ~48% of tropical peat is located, has been estimated at 150 TgCyr$^{-1}$ (Regnier et al., 2013) and 167 TgCyr$^{-1}$, respectively (Sargeant, 2001; Hooijer et al., 2006; Rixen et al., 2021).

Assuming that the eroded mineral and peat soil carbon is remineralised and emitted as $CO_2$ into the atmosphere, it acts as a $CO_2$ source for the climate-active carbon cycle and counts as carbon input like kerogen oxidation. Accordingly, soil erosion should have increased carbon input into the climate-active carbon cycle by 317 TgCyr$^{-1}$ (150 + 167 TgCyr$^{-1}$) to 1,007.0 TgCyr$^{-1}$ (690 + 317 TgCyr$^{-1}$) and decreased the POC burial and therewith the carbon output by 50 TgCyr$^{-1}$ (former POC burial in mineral soils) and ~19.5 TgCyr$^{-1}$ (former peat carbon burial in Indonesia), respectively. Taking additionally the increased POC burial in inland waters and coastal ecosystem of 150 TgCyr$^{-1}$ into account, the carbon output increased by 80.5 TgCyr$^{-1}$ (150 – 50 – 19.5 TgCyr$^{-1}$, Figure 10b). Due to these changes, the pronounced

imbalance between carbon inputs and outputs as seen in the pre-industrial period has almost disappeared. This implies that the degradation of terrestrial ecosystems has reached a level where the associated enhanced supply of lithogenic material and the resulting increased burial of POC in sediments falls below the release of soil carbon. Therefore, it is assumed that the compensatory mechanism that may have contributed to the low variations in atmospheric $CO_2$ concentrations before 1860 becomes inefficient. Nevertheless, the human-induced enhanced supply of lithogenic matter is still assumed to be an important factor that increased the POC burial rate in marine coastal ecosystems.

## Restoration

Ecosystem restoration and expansion are accepted climate change mitigation strategies, because they could increase the carbon storage in biomass and the POC burial; for example, expanding forest cover due to the high carbon storage in trees and restoring degraded ecosystems such as mangroves, whose spatial extent has shrunk by 30–50% since the 1950s (Duarte et al., 2013 and references therein) have been discussed. Bastin et al. (2019) mapped the areas suitable for growing trees (excluding existing trees, croplands and urban areas) and estimated that their expansion could remove around 205 P($=10^{15}$)gC from the atmosphere. However, their study raised doubts concerning, among others, the likely carbon gains from increased forest cover and the space available for more canopy cover (Friedlingstein et al., 2019; Veldman et al., 2019). Hence, the amount of additional carbon that could be stored in trees (40 PgC) and forest soils (2 PgC) was downscaled to about 42 PgC (Veldman et al., 2019). This still represents ~10% of the global living biomass carbon reservoir, with a size of 410–480 Pg C (Bar-On et al., 2018; Spawn et al., 2020).

Mangroves and tropical peatlands were included in this estimate; peat swamp forests were grouped jointly with tropical rain forests in the category of tropical moist broadleaf forests. Mangroves were assumed to contribute ~2.5% (1–1.2 PgC) to the estimated additional carbon storage of 42 TgC with an equal contribution (0.5–0.6 PgC) of biomass and soils. Given a current above-and-below-ground mangrove biomass of ~1.6 PgC (Alongi, 2020), this represents an increase in living mangrove biomass of ~35%. Assuming a recovery time of mangroves equivalent to that of tropical peat swamps (ca. 60–170 years, Hapsari Kartika et al., 2018) suggests an additional POC burial rate of $6.5 \pm 3.5$ TgCyr$^{-1}$. This represents an increase of ~35% of the current figure (19.1 ± 15.8 TgCyr$^{-1}$, see Figure 6), which appears to be achievable due to the restoration of degraded mangroves whose spatial extent has shrunk by approximately 30–50% since the 1950s as mentioned before.

Studies that include tidal marshes and seagrass beds suggest a potential restoration-caused $CO_2$ reduction of 621–1,064 TgCO$_2$e yr$^{-1}$ until 2030 (Pendleton et al., 2012; Griscom et al., 2017; Macreadie et al., 2021). This corresponds to a potential $CO_2$ reduction of $229 \pm 60.4$ TgCyr$^{-1}$, which, assuming a period of 11 years (2019–2030), amounts to a total carbon removal of 2.5 PgC. A $CO_2$ uptake of 2.5 PgC rather than 1 PgC would raise the estimated global ecosystem restoration-caused $CO_2$ uptake to 43.5 PgC, to which coastal benthic ecosystem restoration contributes ~6% (= 2.5 PgC). However, since the current anthropogenic $CO_2$ emission of 10 PgCyr$^{-1}$ (=10,000 TgCyr$^{-1}$) could compensate for $CO_2$ uptake of 43.5 PgC within 4 to 5 years, ecosystem restoration is not an alternative to the reduction of $CO_2$ emissions and the decarbonisation of our economies (Friedlingstein et al., 2019; Friedlingstein

et al., 2020). Nevertheless, ecosystem restoration could be crucially important in achieving climate pledges in countries where potential restoration sites cover a significant area.

However, $CO_2$ sinks in conventional blue carbon ecosystems (Al-Haj and Fulweiler, 2020; Rosentreter et al., 2021; Rosentreter et al., 2023), peat swamps (Günther et al., 2020) and inland waters (Bastviken et al., 2011; Harrison et al., 2021) are also confronted with $CH_4$ and $N_2O$ emissions. Since these are much more potent greenhouse gases than $CO_2$ (Myhre et al., 2013), just a small increase could offset the effects of enhanced $CO_2$ sequestration on the climate (e.g., Günther et al., 2020; Williamson and Gattuso, 2022).

## Future perspectives

Carbon budgets that factor in $CH_4$ and $N_2O$ emissions suffer from a scarcity of data and knowledge gaps, but established methods to quantify stocks and fluxes do exist (IPCC, 2014; The Blue Carbon Initiative, 2014; Needelman et al., 2018). Such is not the case with the quantification of carbon stored by plankton within pelagic coastal ecosystems. Instead of building huge quantities of biomass, carbon fixed by phytoplankton within pelagic ecosystems is exported below the sunlit surface ocean, but only a small part of it is buried in sediment (as was discussed above). The vast majority of the exported biomass is respired and stored as DIC in the ocean. This type of carbon storage is referred to as the biological carbon pump (Volk and Hoffert, 1985). A hypothetical collapse of the biological carbon pump has been assumed to increase atmospheric concentrations of $CO_2$ by 200 to 300 ppm (Heinze et al., 2015), that is, a doubling of the pre-industrial-era figure. It implies a $CO_2$ storage capacity of at least 424–636 PgC, using the widely accepted conversion factor of 2.12 to translate atmospheric $CO_2$ concentrations into PgC (e.g., Wang et al., 2009). Hence, the $CO_2$ storage capacity of phytoplankton seems to be similar to those of living terrestrial plants.

The $CO_2$ uptake efficiency of the biological carbon pump responds to changes in ocean circulation and internal ecosystem processes, for example, particle formation and export as well as carbon-to-nutrient ratios (Heinze et al., 1991; Kwon et al., 2009; Tschumi et al., 2011). However, physical processes (solubility pump) and the respiration of organic matter exported from other coastal and/or terrestrial ecosystems mask the effects of the biological carbon pump on $CO_2$ fluxes across the air–sea interface (Cai, 2011; Bauer et al., 2013; Resplandy et al., 2018; Siddiqui et al., 2023). Although approximately one-third of the POC buried in marine sediment is assumed to originate from terrestrial plants, high $CO_2$ emissions from rivers ($664.5 \pm 181.5$ TgCyr$^{-1}$), and estuaries ($250.0 \pm 125.0$ TgCyr$^{-1}$) (Bauer et al., 2013; Lauerwald et al., 2015) support numerical model (Jacobson et al., 2007; Resplandy et al., 2018; Hauck et al., 2020) and field study results (Najjar et al., 2018; Wit et al., 2018) which show that leached and eroded plant material is largely decomposed on its way towards the ocean (Mathis et al., 2022).

Today, rising atmospheric $CO_2$ concentrations, changing circulation patterns and increasing riverine discharges of nutrients, alkalinity and lithogenic matter as discussed before seem to have transformed the coastal ocean from a $CO_2$ source to a $CO_2$ sink for the atmosphere, with a current average $CO_2$ uptake of about $250 \pm 50$ TgCyr$^{-1}$ (Dai et al., 2022 and references therein). Feedback on the $CO_2$ uptake by the biological carbon pump as well as the PIC and POC burial via the resulting ocean acidification, and the spread of oxygen minimum zones as well as impacts of economical

activates such fisheries, tourisms and seabed mining on coastal carbon cycle (Diaz and Rosenberg, 2008; Altieri et al., 2017; Wit et al., 2018; Sala et al., 2021) have yet to be quantified. This is a crucial step to integrate pelagic ecosystems into the blue carbon concept and to evaluate the sustainability of the rapidly expanding blue economy.

## Summary

Marine coastal ecosystems cover only 5.8% of the Earth's surface but contribute 55.2% to the total carbon transport from the climate-active carbon cycle to the geological carbon cycle via the burial of carbon. Therewith, they play a crucial role as a $CO_2$ sink. The burial of carbon seems to have exceeded the carbon input into the climate-active carbon cycle via weathering and the degassing of $CO_2$ from the interior to the Earth. Land-use changes and the resulting soil degradation and erosion could have initiated this imbalance by increasing the supply of lithogenic matter. Accordingly, land-use change must have been not only a $CO_2$ source for the atmosphere but also a $CO_2$ sink by providing lithogenic material and promoting the burial of POC in coastal marine ecosystems, which may have contributed to the low variations in atmospheric $CO_2$ concentration before 1860.

After 1860, rising $CO_2$ concentrations in the atmosphere suggest that enhanced $CO_2$ emissions caused by land-use changes and the burning of fossil fuel disturbed what was a quasi-steady state. The restoration of ecosystems could mitigate the accumulation of $CO_2$ in the atmosphere by increasing carbon storage in living biomass and the POC burial; benthic coastal ecosystems contribute only 6% to the process. Although this may be of consequence in the context of national carbon budgets, the effects on climate change are still difficult to quantify because of the associated effects of $N_2O$ and $CH_4$. Resolving $CH_4$- and $N_2O$-related issues, reducing uncertainties around carbon fluxes between the climate-active and geological carbon cycles, and the development of suitable methods to quantify changes in the $CO_2$ uptake of the biological carbon pump in pelagic coastal ecosystems are topics that require further research.

**Open peer review.** To view the open peer review materials for this article, please visit http://doi.org/10.1017/cft.2023.20.

**Supplementary material.** The supplementary material for this article can be found at http://doi.org/10.1017/cft.2023.20.

**Acknowledgements.** I would like to thank the editorial team for inviting me to write this article and my colleagues
Nils Moosdorf and Alexandra Nozik from the Leibniz Centre for Tropical Marine Research (ZMT), Bremen, for their support.

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
