## [Reviewer Report]

The abstract needs to be understood on its own, and that is a bit difficult at present. the messages and results can be clearer described, also with sufficient context.

‘’ but also mitigate (and in the past even balanced) their CO 2 emissions by increasing CO 2 storage within marine coastal ecosystems and inland waters’’

Reviewer comment: this is unclear, bu increasing their spatial extent?

‘’ CO 2 storage in ecosystems’’

Reviewer comment: terrestrial?

,’’and warm-water coral reefs are absent.’’

Reviewer comment: what about cold-water coral reefs?

‘’ meadows, are firmly rooted in the ground and store significant amounts of carbon within aboveground biomass. Therefore, they are also considered to be vegetated coastal ecosystems (Fig.’’

Reviewer comment: also a discussion on storage of macroalgaen carbon in deep sea sediments etc, se for ex: https://royalsocietypublishing.org/doi/10.1098/rsbl.2018.0236

‘’ alos foregrounded by sandy beaches’’

Reviewer comment: spelling

‘’ carbonate carbon (PIC = particulate inorganic carbon)’’

Reviewer comment: I would use inroganic carbon as the term here

‘’ carbonate system shifts towards CO 2 and CO 3 2-‘’

Reviewer comment: be consistent on naming, sometimes the chemical formula is used, ie. use carbonate ion or CO3, same for bicarbonate

‘’ Fig. 6 Atmospheric CO 2’’

Reviewer comment: what is meant by “coastal ecosystems” and “steady state” in the graph?

‘’ as do carbon and bicarbonate inputs that equal carbon and bicarbonate outputs. T’’

Reviewer comment: unclear

‘’ and after the blue acceleration of the 1980s,’’

Reviewer comment: define, what is meant by this?

‘’ in well-established blue carbon ecosystems (Macreadie et al., 2017; Mazarrasa et al., 2015’’

Reviewer comment: mostly coral reefs, or general?

‘’geological carbon cycles as explained before.’’

Reviewer comment: refer to section

‘’ input that falls below the output (Fig. 9a). In this case’’

Reviewer comment: these discussions on inputs/outputs is important, but I miss some clearer definitions and introductions of these terms. it should be made clearer in the text how these are defined. it is explained briefly in figure caption to Fig 9, but should also be clearer defined and explained in text

‘’ . To sustain this state, a POC burial rate’’

Reviewer comment: here also the figure caption to Fig 10 should be explained more in the text, perhaps a short method section is needed. how is the steady state calculated? is the “input” in the figure the present-day? perhaps a reference back to Fig 6 and defined time periods is needed? this section is diffucult to follow without some more explanations of methods used to calculate figs 9 and 10

‘’ Nevertheless, ecosystem restoration could be crucially important in achieving climate pledges in countries where potential restoration sites cover a significant area. ‘’

Reviewer comment: there are also discussions on the durability and feasibility of restorations efforts for blue carbon systems, that could be brought in here

Hence, the CO 2 storage capacity of phytoplankton seems to be similar to those of living plants.

‘’ is it meant terrestrial vegetation here?’’

‘’ In the ocean on the shelf approximately 35 – 55% of the introduced terrestrial organic matter is decomposed (Mathis et al., 2022).’’

Reviewer Comment: akward sentence

‘’ associated effects on ecosystems and their functions ‘’

there is also a growing body of litterature of increasing input of terrestrial organic matter to the coast, which could be mentioned here. se for ex https://pubs.acs.org/doi/full/10.1021/acs.estlett.6b00396

---

## [Reviewer Report]

Spatially, the authors discussed the coastal carbon cycle only from a vertical view. However, the coastal carbon cycle is also characterized with horizontal transports of carbon species. See Figure 1 of Mathis et al. (2022) for reference. To avoid misleading, more issues should be involved in this review paper. Moreover, the review paper fully ignored relevant and recent literatures authored by Asian researchers. This is not a balanced style for an international journal submission.

Specific comments and suggestion:

Page 7, the statement of ‘an excess bicarbonate supply favours the CO2 uptake of the ocean by increasing the pH’ is questionable. Because the chemical reaction equation has clearly shown that bicarbonate cannot support the CO2 uptake of seawater. Off some bicarbonate-delivering estuaries in Asia, many of the riverine excess bicarbonate ions are transformed into carbonate ions by biological activities in nearshore areas, which plays the key role in transforming terrestrial carbonate system into seawater carbonate system (Xiong et al., 2019, https://doi.org/10.1029/2019EA000679).

Page 9, as for the CO2 sink/source issue related to the blue carbon and the relevant integrated bicarbonate cycle, the authors may like to refer to the marsh CO2 pump as proposed by Wang and Cai (2004, https://doi.org/10.4319/lo.2004.49.2.0341).

Page 18, the statement of ‘shelf seas operate as CO2 sources in the tropics and CO2 sinks at higher latitudes’ was firstly proposed by Borges et al. (2005, https://doi.org/10.1029/2005GL023053) and questioned by Dai et al. (2013, https://doi.org/10.1002/grl.50390). For example, the Scotian Shelf region (east of Canada) serves as a net CO2 source (Shadwick et al., 2010, https://doi.org/10.5194/bg-7-3851-2010). This is because the seawater is warmed as the cold Labrador Current flows southward, increasing sea surface pCO2 and contributing to CO2 outgassing in the region. However, at approximately the same latitude on the northwest European continental shelf, the Celtic Sea (south of Ireland and west of England) acts as a net CO2 sink (Humphreys et al., 2019, https://doi.org/10.1016/j.pocean.2018.05.001). Its controlling mechanism has been proposed to be an active continental shelf pump process. For this topic, Mathis et al. (2022) did not provide mechanism-based insight with regional details.